ecology/computer modelling and simulation

coral reef, Great Barrier Reef, climate adaptation, climate impacts, coral bleaching, meta-community model

**Author for correspondence:**
Scott A. Condie
e-mail: scott.condie@csiro.au

# Large-scale interventions may delay decline of the Great Barrier Reef

Scott A. Condie[1,2], Kenneth R. N. Anthony[3,4],
Russ C. Babcock[5], Mark E. Baird[1], Roger Beeden[6],
Cameron S. Fletcher[7], Rebecca Gorton[1],
Daniel Harrison[8,9], Alistair J. Hobday[1,2],
Éva E. Plagányi[2,5] and David A. Westcott[7]

[1]CSIRO Oceans and Atmosphere, Hobart, Tasmania, Australia
[2]Centre for Marine Socioecology, University of Tasmania, Hobart, Tasmania, Australia
[3]Australian Institute of Marine Science, Townsville, Queensland, Australia
[4]School of Biological Sciences, The University of Queensland, Brisbane, Queensland, Australia
[5]CSIRO Oceans and Atmosphere, Brisbane, Queensland, Australia
[6]Great Barrier Reef Marine Park Authority, Townsville, Queensland, Australia
[7]CSIRO Land and Water, Atherton, Queensland, Australia
[8]National Marine Science Centre, Southern Cross University, Coffs Harbour, New South Wales, Australia
[9]Marine Studies Centre, School of Geosciences, University of Sydney, Camperdown, New South Wales, Australia

SAC, 0000-0002-5943-014X

On the iconic Great Barrier Reef (GBR), the cumulative impacts of tropical cyclones, marine heatwaves and regular outbreaks of coral-eating crown-of-thorns starfish (CoTS) have severely depleted coral cover. Climate change will further exacerbate this situation over the coming decades unless effective interventions are implemented. Evaluating the efficacy of alternative interventions in a complex system experiencing major cumulative impacts can only be achieved through a systems modelling approach. We have evaluated combinations of interventions using a coral reef meta-community model. The model consisted of a dynamic network of 3753 reefs supporting communities of corals and CoTS connected through ocean larval dispersal, and exposed to changing regimes of tropical cyclones, flood plumes, marine heatwaves and ocean acidification. Interventions included reducing flood plume impacts, expanding control of CoTS populations, stabilizing coral rubble, managing solar radiation and introducing heat-tolerant coral strains. Without intervention, all climate scenarios resulted in precipitous declines in GBR coral cover over the next 50 years. The most effective strategies in delaying decline were combinations that protected coral from both predation

(CoTS control) and thermal stress (solar radiation management) deployed at large scale. Successful implementation could expand opportunities for climate action, natural adaptation and socioeconomic adjustment by at least one to two decades.

## 1. Introduction

The Great Barrier Reef (GBR) is the largest living structure on the planet and currently under intense pressure from climate change and other threats. A crown-of-thorns starfish (CoTS) outbreak began on the GBR in 2010 [1] exacerbating coral mortality associated with a decade of severe tropical cyclones (notably Hamish in 2009, Yasi in 2011 and Debbie in 2017) and successive mass coral bleaching events in 2016, 2017 and 2020 [2,3]. Their cumulative impacts have depleted coral cover to some of the lowest levels in recorded history. The frequency of mass bleaching events and intensity of cyclones are already being influenced by ocean warming [4,5], and ocean acidification is expected to have an increasing impact on coral growth over the next few decades [6]. Beyond these ecological impacts, continued decline of the GBR could jeopardize Australian employment equivalent to 64 000 full-time jobs and economic value of AU$6.4 billion per annum [7].

Effective global climate change mitigation is clearly essential to the future of the GBR [8]. In addition, a variety of strategies have been suggested to offset future impacts on the reef. Shorter-term options have primarily focused on enhancing ecosystem resilience and adaptive capacity by improving water quality [9–11] or ensuring compliance within protected areas [12]. However, there is increasing acceptance of the need for more targeted interventions. An existing programme is the direct eradication of CoTS using lethal injection, which started targeting high priority reefs in 2011 [13] and in 2019 expanded to five control vessels operating over much of the GBR [14]. A wider range of interventions may become technically feasible and socially acceptable if the GBR declines further. Here, we consider a diverse range of interventions, including some that are yet to be tested *in situ* or at large scale.

The effectiveness of any intervention in protecting the GBR will depend on many system interactions and may only become apparent over multi-decadal timescales. Gaining insights into these interactions, with a view to identifying practical interventions, initially requires a systems modelling approach that can be used to formulate and test alternative strategies [15–17]. Ultimately, such models may be used to plan adaptation pathways involving multiple interventions applied adaptively under evolving technological, environmental and social conditions [18].

## 2. Material and methods

GBR interventions likely to be technically feasible and cost-effective within the near future have previously been identified (table 1). We have tested these strategies within a reef meta-community model describing key physical and biological processes operating on coral reef systems exposed to tropical cyclones, impacts from flood plumes, mass bleaching events and ocean acidification (figure 1). The Coral Community Network (CoCoNet) model has previously been implemented on a generic network of reefs [17]. The current implementation on the GBR resolved 3753 individual reefs with enhanced representation of coral communities, as well as a range of human interventions aimed at protecting or restoring coral communities (figure 1). Key components of the model workflow are shown in figure 2 and key assumptions are listed in table 2, with more detailed descriptions provided below.

The CoCoNet model has been calibrated against the Australian Institute of Marine Science (AIMS) Long-Term Monitoring Program (LTMP) dataset [55] at both the individual reef scale [28,56] and reef network scale [17]. It has also undergone continuous qualitative evaluation by the Reef Restoration and Adaption Program (RRAP) group of 33 experts (https://www.gbrrestoration.org/about-us) to confirm it was fit for purpose. Results demonstrate realistic simulations of coral cover and CoTS outbreak densities, and emergent system responses such as CoTS outbreaks and coral recovery at close to their observed periodicity [17]. Here, we further demonstrate the model's ability to reproduce observed coral cover trends under realistic environmental forcing conditions.

### 2.1. Population dynamics

Each reef had a fixed coral-carrying capacity proportional to the area of the reef. Coral communities consisted of five coral groups whose species are relatively abundant on the GBR [57]. These groups

**Table 1.** Definition of intervention strategies (IS) targeting flood plumes, CoTS or corals (italicized) identified through the Great Barrier Reef Blueprint for Resilience (http://elibrary.gbrmpa.gov.au/jspui/handle/11017/3287) and the Reef Restoration and Adaptation Program (https://www.gbrrestoration.org/interventions). All on-reef actions targeted a draft set of 289 priority reefs identified by GBR management agencies, with CoTS control then extending to other reefs as vessel capacity allowed.

| no. | strategy description | examples of actions | model implementation |
| --- | --- | --- | --- |
| IS0 | no intervention | no actions | all interventions turned off |
| IS1 | reduce the impacts of *flood plumes* and control *CoTS* control at currently planned levels | reduce sediment and nutrient run-off [9]; existing *CoTS* control programme [13,19] | over a 20-year timescale, gradually restrict plume footprints (manifested as reduced coral growth rates and enhanced *CoTS* recruitment) on the inner reef. In the model, this was equivalent to reducing the flood impacts (but not direct wave impacts) of cyclones by one cyclone category [17]. reduce *CoTS* populations below an ecological threshold (equation (2.17)) starting with priority reefs and then other reefs in random order. The number of reefs treated per annum was limited by the number of control vessels (five). |
| IS2 | accelerate reduced impacts of *flood plumes* | rapidly reduce sediment and nutrient run-off [9] | catchment restoration from IS1 with implementation time reduced by 80% to 4 years |
| IS3 | enhance *CoTS* control programme | add more *CoTS* control vessels [13,14] | CoTS control from IS1 with twice as many control vessels (i.e. 10 vessels) |
| IS4 | increase *coral* recruitment by stabilizing coral rubble following cyclone and mass bleaching events | remove or bond coral rubble using physical, chemical or biological approaches | reduce timescale for substrate consolidation following cyclone and mass bleaching events from 5.5 to 2 years on priority reefs with coral cover <20%, up to a maximum of 100 ha of reef |
| IS5 | protect *coral* from mass bleaching by shading or cooling priority reefs | add reflective water surface films; pump cooler water from depth | reduce heat exposure by 12 degree heating weeks (DHW) on all priority reefs |
| IS6 | protect *coral* from mass bleaching using regional-scale shading | artificially generate clouds (cloud-brightening) during periods of bleaching risk | reduce probability of mass bleaching mortality by decreasing annual exposure by 4 DHW on all reefs |

*(Continued.)*

**Table 1.** (*Continued.*)

| no. | strategy description | examples of actions | model implementation |
| --- | --- | --- | --- |
| IS7 | introduce thermally tolerant *coral* strains [20] | release coral larvae or outplant coral fragments [21] | add 10 ha of thermally tolerant (staghorn) corals per annum to priority reefs with <20% coral cover. Allow interbreeding with 10% of existing staghorn corals with 50% of offspring retaining enhanced thermal tolerance |
| IS8 | protect *corals* from ocean acidification using chemical or biological processes | add chemical treatments [22] or establish macroalgae farms near reefs [23] | halt decline in coral growth rates on all priority reefs |
| IS3&6 | enhance *CoTS* control and protect *coral* using regional-scale shading | as described for IS3 and IS6 | combine IS3 and IS6 |
| IS3&7 | enhance *CoTS* control and introduce thermally tolerant *coral* strains | as described for IS3 and IS7 | combine IS3 and IS7 |
| IS6&7 | protect *coral* using regional-scale shading and introduce thermally tolerant *coral* strains | as described for IS6 and IS7 | combine IS6 and IS7 |
| IS3&6&7 | enhance *CoTS* control, protect *coral* using regional-scale shading and introduce thermally tolerant *coral* strains | as described for IS3, IS6 and IS7 | combine IS3, IS6 and IS7 |

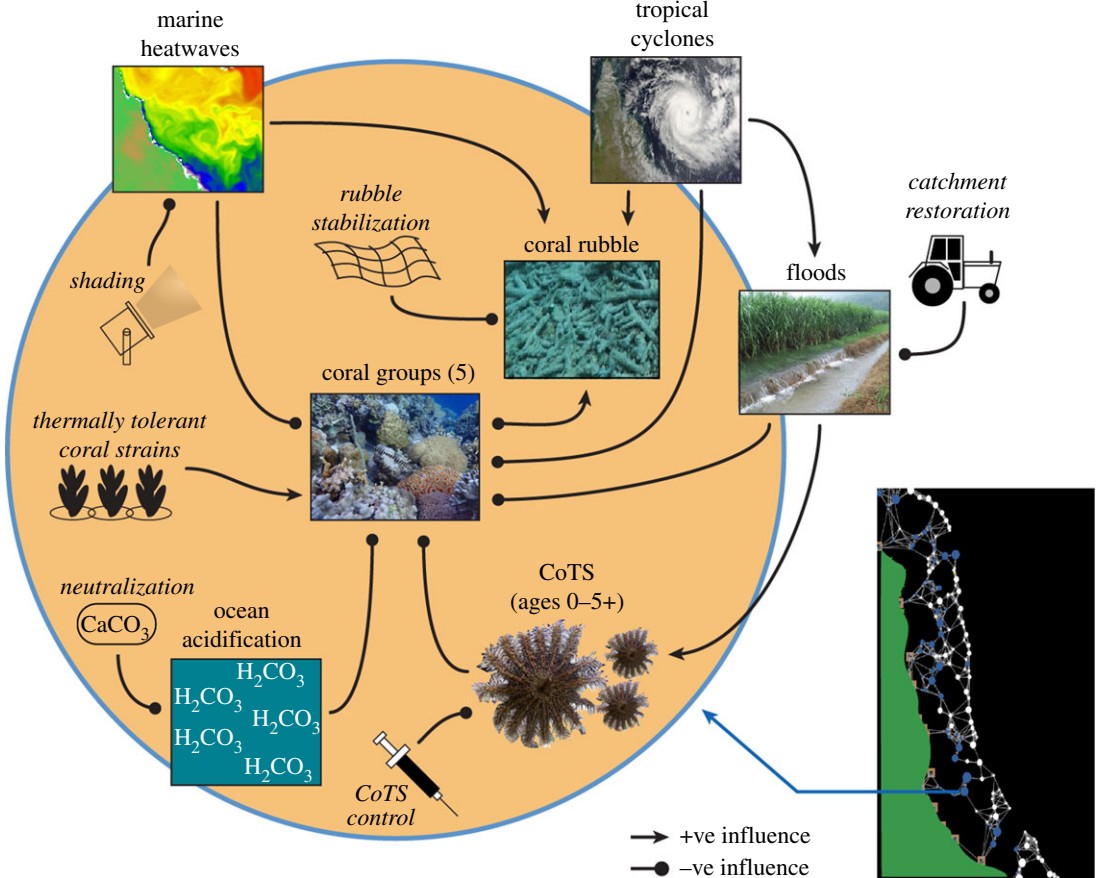

**Figure 1.** Components of the Coral Community Network (CoCoNet) model showing within-reef interactions on the left (interventions shown as line drawings) and between-reef interactions on the lower right.

nominally corresponded to staghorn *Acropora*, tabular *Acropora*, *Montipora*, *Poritidae* and favids, distinguished within the model in terms of their growth rates, preference by CoTS, and susceptibility to environmental impacts such as cyclones and marine heatwaves (tables 3–5). Differences in fecundity among coral groups were assumed to be negligible compared with differences in environmental susceptibility [60] and independent of geographical location along the GBR [61] (assumption 1a).

CoTS populations were size-structured, differentiating larvae (age 0 years), herbivorous juveniles (age 1 year) and four corallivorous adult classes (ages 2, 3, 4 and 5+ years). By directly equating size with age, the potential for delayed transition from juvenile to adult stages [25] was excluded (assumption 1b). Trophic interactions between corals and CoTS were calculated using a formulation (equations (2.9)–(2.11)) that included doubling of adult CoTS predation rates until age 4 [26] when they began to move into a senescent phase [1,62] (assumption 1c). CoTS had a preference for faster-growing corals [27,28] (assumption 1d) and populations declined when these became rare (equations (2.2), (2.3) and (2.12)) (assumption 1e). Rate parameters such as growth, predation and natural mortalities were fitted to data from the AIMS LTMP [28,56].

## 2.2. Reef connectivity

Reef connectivity was determined by spawning, larval transport by ocean currents and successful settlement onto either a natal reef (self-recruitment) or neighbouring reefs (cross-recruitment). For all coral groups, larval production was proportional to their area coverage (assumption 2a). CoTS larval production was proportional to the number of adult starfish (assumption 2a) and also increased by a factor of 4 for each age class [35] before plateauing after age 4 years [1,62] (assumption 2c).

Larval transport was modelled as directed links that appeared and disappeared from year to year with exchanges that also varied stochastically to represent variability in ocean currents and larval survival. A maximum travel distance for coral larvae was set at 90 km [33,63] and for CoTS at 150 km [31,64,65]. The probability of successful recruitment onto a reef from spawning on an upstream reef

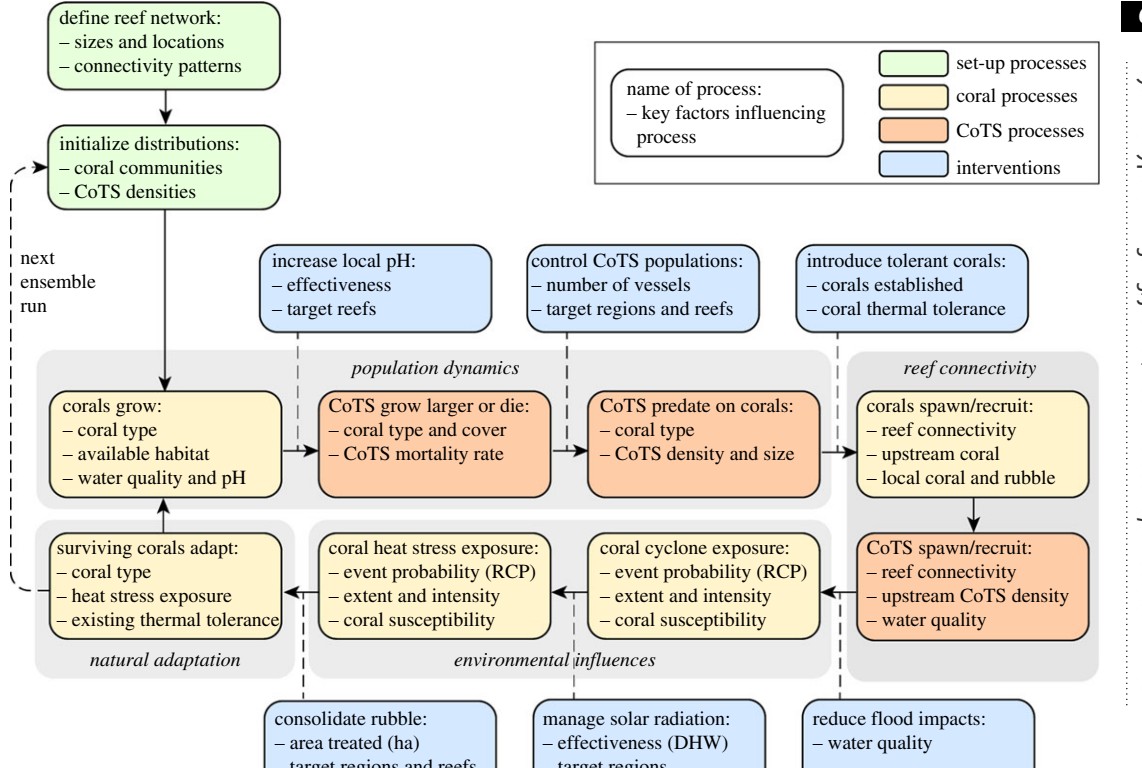

**Figure 2.** Key components and workflow for the CoCoNet model, including model initialization, population dynamics for corals and CoTS, spawning and reef connectivity, environmental influences, natural adaptation of corals and six types of intervention applied either individually or in combination.

was proportional to the connectivity estimated from particle tracking model results, although recruitment to a reef was reduced by the proportion of coral rubble previously generated by cyclone and bleaching-induced coral mortality (assumption 2b). While particle tracking provided a direct estimate of relative connectivity, because larvae cannot be tracked over large spatial scales and larval survival rates are uncertain, the proportionality constant between relative connectivity and successful recruitment could not be determined empirically. It was, therefore, treated as a calibration parameter independently for coral and CoTS connectivity. Coral and CoTS populations were found to be particularly sensitive to the CoTS connectivity parameter [17].

The structure of the reef connectivity networks for corals and CoTS were estimated using ocean current patterns from the eReefs 1 km resolution hydrodynamic model [30–32,66,67]. Both coral and CoTS spawning events were simulated over 3 years of available currents (2016–2018) by releasing particles from all reefs over their respective spawning periods. Particles were advected by the current fields, with the imposition of biological constraints such as preferred swimming depths and larval mortality rates. Broad-scale geographical differences in larval mortality were also modelled, with temperature preferences for CoTS larvae [36] assumed to peak around 15–16°S where outbreaks typically initiate [1] (assumption 2d). A directed link was established between two reefs when a particle released from one passed within 1 km of another during the period when larvae would have been competent to settle. The 1 km 'capture halo' allowed for any directed swimming of larvae (weak for corals and CoTS) and the limited resolution of current fields that may not fully resolve features such as lee eddies. All directed links were combined into a reef connectivity matrix with each element indicating the number of particle connections between two reefs. Connectivity matrices were generated for each of the three coral spawning seasons and each of the three CoTS spawning seasons.

Using the connectivity matrices directly in CoCoNet would have not only restricted the choice of connectivity patterns to the three modelled spawning periods available through eReefs, but would also have reduced the computational speed of the model by several orders of magnitude. Instead, links between reefs were made each year with a probability that depended on direction, distance and the median weighted in-degree centrality (figure 3a). Weighted in-degree centrality is the product of the number of incoming connections and the average weight of those connections [68] and was

**Table 2.** Key assumptions and associated limitations. The simplest assumption consistent with empirical data was adopted, recognizing that the data inevitably have limited spatial, temporal and/or species coverage. Further details are provided in the text.

| model component | assumption | limitations and references |
|---|---|---|
| 1. population dynamics | a. intrinsic growth rates of corals and CoTS are relatively uniform over the GBR | while the linear extension of corals has been found to decline with latitude, increases in coral density appear to be relatively uniform [24] |
| | b. CoTS grow to a larger size class each year | when their coral prey is not readily available, herbivorous juvenile CoTS may delay maturing to corallivorous adults [25], which could influence both outbreak patterns and the effectiveness of CoTS control |
| | c. predation rates by adult CoTS double with each size class, plateauing at age 4 years (equations (2.9)–(2.11)) | [26] |
| | d. CoTS have a preference for faster-growing corals (e.g. *Acropora*) over slower-growing corals (e.g. *Portidae*) (equations (2.9)–(2.11)) | [27,28] |
| | e. adult CoTS mortality rates increase when coral is scarce (equation (2.12)) | [29] |
| 2. reef connectivity | a. recruitment is proportional to both the density of spawners on upstream reefs and hydrodynamic connectivity that accounts for coral and CoTS larval behaviour (equations (2.1) and (2.4)). The proportionality constant was estimated as part of the model calibration process | [30–33] there are many factors that can potentially confound relationships between hydrodynamic connectivity, larval supply, larval settlement and larval survival [34] |
| | b. coral recruitment is inhibited by coral rubble generated by previous cyclone and bleaching events. Whereas coral rubble provides suitable habitat for CoTS recruitment | [29] |
| | c. contributions of adult CoTS to spawning and recruitment increases by a factor of 4 with each successive size class, plateauing at age 4 years (equation (2.1)) | [35] |
| | d. CoTS recruitment is dependent on latitude (i.e. temperature) with optimal conditions around 15° S, where outbreaks typically initiate | [36] |

(*Continued.*)

**Table 2.** (*Continued.*)

| model component | assumption | limitations and references |
| --- | --- | --- |
| 3. environmental influences | a. the statistical frequency and intensity of heatwaves and intense cyclones increases in the future at rates dependent on the RCP scenario (figure 4b,c) | uncertainties associated with downscaling of climate projections [37,38] have been partially captured by using an ensemble modelling approach |
| | b. coral mortality during cyclones and bleaching events is dependent on cyclone category and degree heating weeks respectively, as well as coral type (figure 4a,d) | model parametrization relied on quantitative results from a relatively small number of empirical studies [39–41] |
| | c. CoTS recruitment is enhanced on inner-shelf reefs by flood plumes (equation (2.1)) | while the significance of this link continues to be contested in the literature [42–46], it had only a small effect at the scale of the GBR |
| | d. acidification reduces coral growth rates (equation (2.16)), but does not effect CoTS growth rates | [47,48] laboratory studies indicate that acidification has a negative effect on CoTS larvae [49] and a positive effect on the growth of juveniles (via soft tissue growth) [50]. However, the net effect over their life history is uncertain. |
| 4. adaptation | a. when adaptation is modelled, corals surviving thermal heatwaves are allocated enhanced thermal tolerance (equation (2.15)) with an associated growth rate penalty (equation (2.16)). In the absence of further heat stress, these characteristics return to their intrinsic values over timescales of 10 years for the fastest-growing corals, to 100 years for the slowest growing corals (equation (2.16)) | while the parametrization is broadly consistent with available empirical data [51], this aspect of the model has high uncertainty and will require ongoing study and refinement |
| | b. changes in the thermal tolerances of corals are primarily determined by local adaptation to stress (i.e. thermal tolerance is not a dominant trait) | while traits could propagate from reef to reef, averaging at the reef scale tended to dilute their influence |
| 5. flood plume mitigation | a. maximum improvement in flood plume impacts is approximately 42% of the difference between southern and far northern catchments on the GBR and can be achieved by 2040 | condition of the far northern catchments has been used previously as an indication of the maximum possible improvement that might be achievable through catchment restoration [52] |

(*Continued.*)

**Table 2.** (*Continued*.)

| model component | assumption | limitations and references |
| --- | --- | --- |
| 6. CoTS control | a. the efficiency of CoTS control (per vessel) remains unchanged into the future | large improvements in the efficiency of CoTS control may be achievable through improvements in monitoring and/or detection, or through introduction of biological controls [1,14] |
| | b. the detectability of CoTS by divers is 37% for small adults (age 2 years) and increases with age until plateauing from age 4 years | [53] |
| 7. rubble stabilization | a. rubble stabilization can be deployed at scales of approximately 100 ha yr$^{-1}$ | while existing techniques may be scalable given sufficient resources, this scenario is orders of magnitude larger than any past deployment |
| 8. shading | a. local solar radiation management can reduce heating on individual reefs by up to 12 DHW | for some reefs, heat reduction may be limited by warmer water flowing onto the reef |
| | b. solar radiation management can reduce heating by 4 DHW across the entire GBR | large-scale solar radiation management technology is in early stages of development and testing [54]. Major uncertainties in its efficacy and cost-effectiveness remain |
| 9. thermally tolerant corals | a. thermally tolerant coral strains can be bred and deployed with coverages of approximately 10 ha yr$^{-1}$ | while existing techniques may be scalable given sufficient resources, this scenario is orders of magnitude larger than any past deployments |
| | b. thermally tolerant corals can interbreed with up to 10% of existing staghorn corals, with 50% of offspring retaining enhanced thermal tolerance | the potential for hybridization and resulting levels of thermal tolerance will be strongly dependent on the species used. This scenario is indicative only |

**Table 3.** Model equations relating to age-structured CoTS populations and coral functional groups (sa, staghorn *Acropora*; ta, tabular *Acropora*; tt, thermally tolerant; mo, *Montipora*; po, *Poritidae*; fa, favids).

| description | equation | no. |
|---|---|---|
| *CoTS population dynamics* | | |
| CoTS age 0 | $N_{y+1,0} = \sum_{\text{reefs}} \left( f_{y+1}^{CoTS} \dot{N}_{y+1,2} + 4\, f_{y+1}^{CoTS} \dot{N}_{y+1,3} + 16\, f_{y+1}^{CoTS} \dot{N}_{y+1,4} + 16\, f_{y+1}^{CoTS} \dot{N}_{y+1,5} \right)$ | (2.1) |
| CoTS age $a = 1:4$ | $N_{y+1,a} = N_{y,a-1} e^{-f(C_y) M_{y,a}^{CoTS}} - N_{y,a} H_{y,a}^{CoTS}$ | (2.2) |
| CoTS age 5+ | $N_{y+1,5+} = (N_{y,4} + N_{y,5+}) e^{-f(C_y) M_{y,5+}^{CoTS}} - N_{y,5+} H_{y,5+}^{CoTS}$ | (2.3) |
| *coral population dynamics* | | |
| coral groups | $C_{y+1}^g = C_y^g(1 + r_y^g(1 - C_y^g/K) - Q_y^g - M_y^{g,C_yc} - M_y^{g,Ble}) + \sum_{\text{reefs}} f_y^g C_y^{,g}$   ($g =$ sa, ta, tt, mo, fa, po) | (2.4) |
| coral groups preferred by CoTS | $C_y^f = C_y^{sa} + C_y^{ta} + C_y^{tt} + C_y^{mo}$ | (2.5) |
| coral groups not preferred by CoTS | $C_y^m = C_y^{po} + C_y^{fa}$ | (2.6) |
| coral cover fraction | $C_y^c = (C_y^f + C_y^m)/K$ | (2.7) |
| coral diversity (evenness index) | $\mathcal{J} = \dfrac{-(C_y^{sa} \ln(C_y^{sa}/K) + C_y^{ta} \ln(C_y^{ta}/K) + C_y^{mo} \ln(C_y^{mo}/K) + C_y^{po} \ln(C_y^{po}/K) + C_y^{fa} \ln(C_y^{fa}/K))}{K \ln(5)}$ | (2.8) |
| *terms for CoTS predation on coral* | | |
| coral groups preferred by CoTS | $Q_y^g = (1 - \rho_y) \dfrac{p_1(0.2 N_{y,1} + N_{y,2} + 2 N_{y,3} + 4 N_{y,4} + 4 N_{y,5})}{1 + e^{-(0.2 N_{y,1} + N_{y,2} + 2 N_{y,3} + 4 N_{y,4} + 4 N_{y,5})/p_2}}$   ($g =$ sa, ta, tt, mo) | (2.9) |
| coral groups not preferred by CoTS | $Q_y^g = \rho_y \dfrac{p_1(0.2 N_{y,1} + N_{y,2} + 2 N_{y,3} + 4 N_{y,4} + 4 N_{y,5})}{1 + e^{-(0.2 N_{y,1} + N_{y,2} + 2 N_{y,3} + 4 N_{y,4} + 4 N_{y,5})/p_2}}$   ($g =$ fa, po) | (2.10) |
| switch function | $\rho_y = e^{-C_y^f/C_y^m}$ | (2.11) |

(*Continued.*)

**Table 3.** (*Continued.*)

| description | equation | no. |
|---|---|---|
| coral influence on COTS mortality | $f(C_y^f) = 1 - \tilde{p}\,\dfrac{C_y^f}{1 + C_y^f}$ | (2.12) |
| *terms for environmental effects on coral (g = sa, ta, tt, mo, fa, po)* | | |
| bleaching-induced coral mortality | $M_y^{g,Ble} = 1 - e^{\left(-0.1e^{\max(0, DHW_y - S_y - T_y^g)}\right)}$ | (2.13) |
| intrinsic coral thermal tolerance | $\mathcal{T}_o^g = 3.5 - 5_o^g$ | (2.14) |
| coral thermal tolerance following bleaching | $\mathcal{T}_{y+1}^g = \min(\mathcal{T}_y^g(1+\mathcal{A})^{M_y^{g,Ble}},\ \mathcal{T}_o^g + \mathcal{P})$ | (2.15) |
| influence of thermal adaptation and ocean acidification on growth | $r_y^g = (r_o^g(1 - 0.01(\mathcal{T}_{y+1}^g - \mathcal{T}_o^g)))^{(1+k(1-p^{OA})(RCP \cdot r_o^g/r_o^{sa})^{0.5})}$ | (2.16) |
| *CoTS control* | | |
| ecological threshold (for coral decline) | $N_{y,ecol} = \alpha(20(C_y^f/K) + 4)$ | (2.17) |
| control dives (to reach ecological threshold) | $\mathcal{D}_y = 4.18\left(\dfrac{N_{y,2} + N_{y,3} + N_{y,4} + N_{y,5+}}{\alpha}\right)^{0.667}$ | (2.18) |
| *ensemble statistics* | | |
| Cohen's *d* comparing ensemble *i* with the baseline ensemble *b* for year *y* | $d_y = \dfrac{\bar{C}_{yi}^c - \bar{C}_{yb}^c}{s_{yb}}$ | (2.19) |

**Table 4.** Definitions of model variables.

| variable | definition |
| --- | --- |
| CoTS | (age: $a = 0, 1, 2, 3, 4, 5+$) |
| $N_{y,a}$ | number of CoTS of age $a$ at the start of year $y$ |
| $\tilde{N}_{y,a}$ | number of CoTS of age $a$ on connected reefs in year $y$ |
| $H^{CoTS}_{y,a}$ | fraction of CoTS of age $a$ removed through control programmes during year $y$ |
| coral groups | (group: $g$ = sa, ta, mo, tt, fa, po) |
| $C^g_y$ | cover of coral group $g$ at the start of year $y$ |
| $\tilde{C}^g_y$ | cover of coral group $g$ on connected reefs in year $y$ |
| $Q^g_y$ | cover fraction of coral group $g$ consumed by COTS during year $y$ |
| $M^{g,Cyc}_y$ | cyclone-induced mortality of coral group $g$ in year $y$ |
| $\mathcal{T}^g_y$ | thermal tolerance (in DHW) of coral group $g$ in year $y$ |
| environmental conditions | |
| $DHW_y$ | degree heating weeks at a reef over year $y$ |
| human interventions | |
| $\mathcal{S}_y$ | effect of artificial shading or cooling (in DHW) at a reef over year $y$ |
| $p^{OA}$ | level of artificial protection from ocean acidification [0 1] |
| ensemble statistics | |
| $\bar{C}^c_{yi}$ | ensemble average of average coral cover fraction at the start of year $y$ for ensemble $i$ |
| $s_{yi}$ | ensemble standard deviation in average coral cover fraction at the start of year $y$ for ensemble $i$ |

computed from the connectivity matrices. Its median value was calculated for every $0.2 \times 0.2°$ cell using the 3 years of data for both corals and CoTS (figure 3b). A third-order (cubic) polynomial surface in longitude, latitude and weighted in-degree was then fitted on the same geographical grid using linear regression. Third-order polynomials were found to capture the broad-scale variations in in-degree across the GBR with much lower RMS errors than could be achieved with a second-order (quadratic) polynomial.

The cubic surface provided a connectivity probability distribution (CPD) for the reef network. For each spawning event, the probability of forming an incoming link to any reef increased in proportion to the CPD. There was also preferential linking [69] of larger reefs to reflect their larger capture halos. This process gave the network a scale-free structure with larger reefs tending to form connectivity hubs, consistent with previous graph theory analysis targeting one section of the GBR [70].

The final connectivity network provided relative probabilities of links forming between any two reefs. However, recruitment of larvae to any reef will ultimately be influenced by a range of survival factors that cannot be measured directly or inferred from the limited genetic data currently available for the GBR. The mean number of links and mean recruitment for each coral and CoTS group were, therefore, estimated through a calibration process aligning coral and CoTS population trends with observations from the LTMP (described below).

## 2.3. Environmental influences

Reefs were subjected to environmental stressors in the form of tropical cyclones and flood plumes, heatwaves and ocean acidification. These stressors changed over the simulations on the basis of historical data prior to 2020, and then according to statistical climate projections (assumption 3a). These projections corresponded to Representative Concentration Pathways RCP 2.6, 4.5 and 8.5 [71], noting that the ocean heating effects of RCP 4.5 and RCP 6.0 are very similar to 2070 [72]. The resulting scenarios (described below) are considered plausible and consistent with published estimates. However, the large uncertainties inherent in both the modelled climate projections and their impacts ultimately necessitated the use of an ensemble modelling approach (described below).

**Table 5.** Model parameter values, both fixed inputs and estimated by fitting to the LTMP data. In instances where ranges are given, parameters were varied randomly within that range throughout model runs.

| parameter | description | value/range | estimation method | reference |
| --- | --- | --- | --- | --- |
| *CoTS* | | | | |
| $M^{COTS}$ | predation and natural mortality | 2.41–2.71 | fitted to LTMP | [28] |
| $\tilde{p}$ | effect of coral on COTS mortality | 0.10–0.82 | fitted to LTMP | [56] |
| $f_y^{CoTS}$ | recruitment per COTS from connected reefs in year y | 0–1000 | fitted to LTMP | [17] |
| $\alpha$ | conversion factor: control programme CoTS ha$^{-1}$ to CoTS per manta tow | 0.015 | fitted to LTMP | [58] |
| *coral groups* | | | | |
| $r_o^{sa}$ | intrinsic growth rate in year y | 0.5 yr$^{-1}$ in 1950 | pre-specified | [28] |
| $r_o^{ta}$ | intrinsic growth rate in year y | 0.4 yr$^{-1}$ in 1950 | pre-specified | [40] |
| $r_o^{tt}$ | intrinsic growth rate in year y | 0.4 yr$^{-1}$ in 1950 | pre-specified | [40] |
| $r_o^{mo}$ | intrinsic growth rate in year y | 0.3 yr$^{-1}$ in 1950 | pre-specified | [40] |
| $r_o^{po}$ | intrinsic growth rate in year y | 0.1 yr$^{-1}$ in 1950 | pre-specified | [28] |
| $r_o^{fa}$ | intrinsic growth rate in year y | 0.05 yr$^{-1}$ in 1950 | pre-specified | [40] |
| $p_1$ | effect of COTS on coral | 0.0–0.2 | fitted to LTMP | [56] |
| $p_2$ | effect of COTS on coral | 9 | pre-specified | [28] |
| $f_y^{lg}$ | recruitment of coral group g from connected reefs in year y | 0–2 × 10$^{-4}$ | fitted to LTMP | [17] |
| $K$ | maximum potential coral habitat available on a reef | GBR average 3000 | pre-specified (arbitrary units) | [28] |
| *environmental effects on corals* | | | | |
| RCP | climate scenario specification | 2.6, 4.5, 8.5 | pre-specified | |
| $\mathcal{A}$ | adaptability of corals to thermal stress | 0, 5 | pre-specified | |
| $\mathcal{P}$ | maximum thermal plasticity of corals | 12 DHW | pre-specified | |
| $k$ | factor controlling annual decline in coral growth due to ocean acidification | 0.0024 | fitted to observed growth rates | [59] |
| *ensemble statistics* | | | | |
| $n$ | number of model runs within ensemble | 100 | pre-specified | |

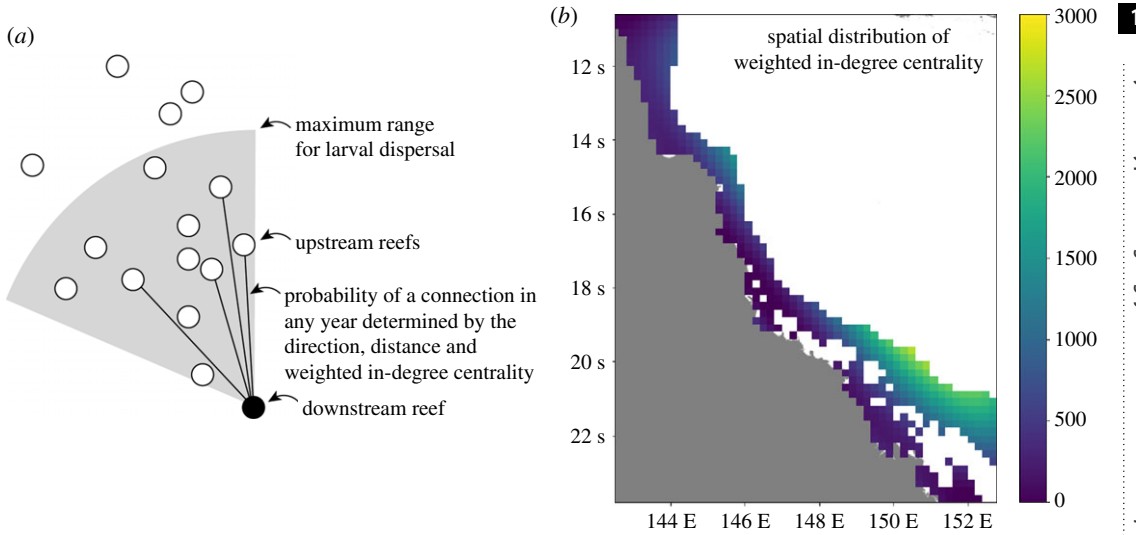

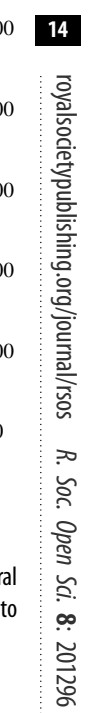

**Figure 3.** (*a*) Factors controlling connections to a downstream reef. (*b*) In-degree centrality of reefs (averaged over three coral spawning periods: 2016–2018) mapped onto a 0.2 × 0.2° grid. In-degree centrality values ranged from 0 (dark purple) to 2638 (yellow) with an average of 676. High values in the southeast reflect high densities of small interconnected reefs.

## 2.4. Tropical cyclones and flood plumes

Parametrization of tropical cyclones (including tropical lows) were represented as described previously [17]. Cyclone events were applied stochastically at frequencies and intensities consistent with recent historical conditions [73–75]. Coral mortalities within the spatial footprint of different cyclone intensities (categories 1–5) were parametrized using results from post-cyclone surveys [17,39] (assumption 3b). However, these too were applied randomly among individual reefs so as to capture the high spatial variability in mortality that is typically observed [76]. There was also a commensurate increase in coral rubble cover at each reef impacted by a cyclone.

Three levels of susceptibility to cyclone damage were specified (figure 4*a*). The two fastest-growing coral groups (staghorn *Acropora* and tabular *Acropora*) had the same high susceptibility to cyclone damage and the two slower-growing coral groups (*Faviidae* and *Poritidae*) had the same lower susceptibility, with *Montipora* midway between these levels. Differentiating only three levels of susceptibility reflects variable morphology within each coral group and is consistent with the relatively coarse levels of differentiation identified empirically [39,80,81].

Cyclone-induced flood plumes also reduced coral growth rates and increased rates of CoTS recruitment (assumption 3c). Coral growth decayed exponentially from its offshore value towards zero at the coastline due to factors such as elevated nutrients and turbidity [52,82], whereas CoTS recruitment increased exponentially towards the coast peaking at five times the offshore value at the coastline [11,46,83]. Throughout the GBR, the offshore scale of these distributions also increased with increasing cyclone category. The maximum offshore exponential scale was limited to 75 km, consistent with the estimated influence of river flood plumes [84] and scales for offshore transport of fine sediments [85].

Future projections assumed that while the frequency of category 1–3 cyclones remained unchanged, it could increase by up to 21% for category 4 cyclones and 42% for category 5, depending on the climate scenario (assumption 3a). These values fall within the range of recently reviewed estimates for the South Pacific from climate simulations [5,86–88] and extrapolation of historical trends [89]. The resulting projected frequency of category 5 cyclones after 2050 was still less than the frequency observed on the GBR over the past decade (2010–2019) and, therefore, not beyond the realistic range. Remaining uncertainties in cyclone frequency tended to become less significant over time as heatwave impacts began to dominate projected coral mortalities.

## 2.5. Marine heatwaves

Similar to tropical cyclones, marine heatwaves were implemented as random events dependent on the cumulative exposure of reefs to high temperatures. This exposure is expressed in terms of degree heating weeks (DHW), acknowledging that bleaching levels are usually influenced by a combination

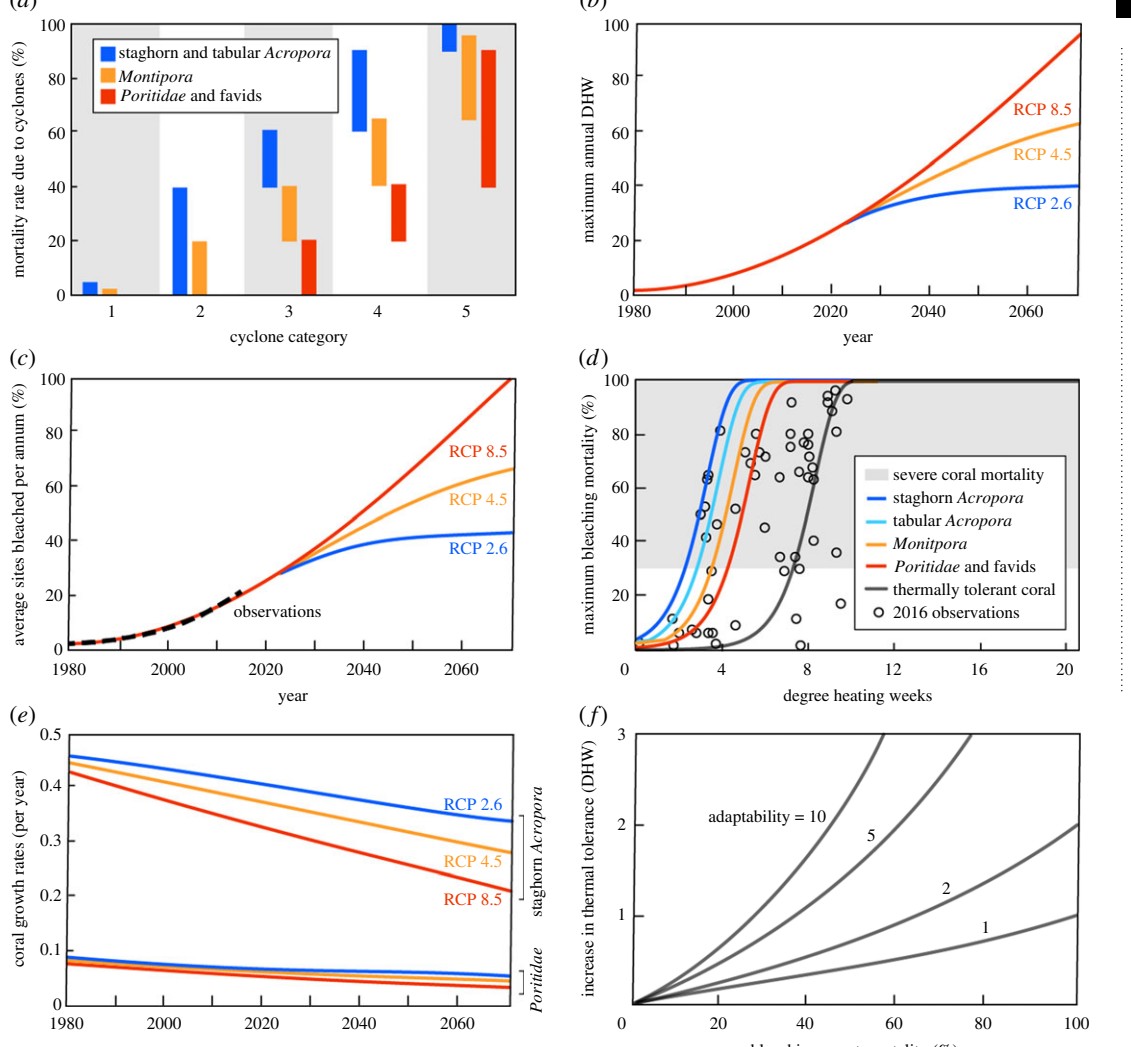

**Figure 4.** (*a*) Ranges of mortality experienced by corals within the impact zone for each cyclone category [17,73,77]. (*b*) Maximum annual DHW used under the three RCP scenarios [71]. Each year, DHW were set at a level randomly selected from below the maximum annual DHW curve. (c) Average proportion of locations bleached per annum under the three RCP scenarios and corresponding estimates from empirical data for 1980–2016 [41]. The long-term values are consistent with the frequency of bleaching (greater than 2° heating months) estimated from climate model projections for RCP 2.6 (0.35–0.45) and RCP 4.5 (0.55–0.75) [4,78], as well as forecasts of annual bleaching across nearly all of the GBR by 2070 under RCP 8.5 [79]. (*d*) Maximum bleaching mortality as a function of DHW for each of the coral groups (equation (2.13)), including the thermally tolerant strain of staghorn *Acropora*. Also shown are observed bleaching mortality rates on individual reefs following the 2016 bleaching event on the GBR [40]. (*e*) Modelled decline in coral growth rate for fast-growing staghorn *Acropora* and slow-growing *Poritidae* due to ocean acidification (equation (2.16)). These trends exclude any effects of natural adaptation. (*f*) Increase in thermal tolerance of coral surviving a bleaching event as a function of bleaching mortality for a range of adaptability levels (equation (2.15)). Initial thermal tolerance values were: 1.0 DHW for staghorn *Acropora*; 1.5 DHW for tabular *Acropora*; 2.0 DHW for *Montipora*; 3.0 DHW for *Poritidae* and favids and 6.0 DHW for thermally tolerant corals (equation (2.14)).

of water temperature and irradiance [90]. Coral bleaching mortality within the spatial footprint of marine heatwaves increased with DHW (assumption 3b), with a commensurate increase in coral rubble cover the following year. Prior to 2020 heatwaves followed observed patterns, after which plausible future scenarios for the maximum annual DHW were described by a sigmoid curve (hyperbolic tangent) with parameters estimated from past sea surface temperature distributions [37] and extrapolation of past bleaching events [71] (figure 4*b*).

In any year, a marine heatwave could occur with the maximum geographical extent of the bleaching event increasing with the maximum annual DHW. The ratio of these two quantities was set to ensure that

the average proportion of locations bleached on the GBR per annum aligned with corresponding empirical estimates from the Australasian region (figure 4c) [41].

With size and DHW distributions for marine heatwaves aligned with available empirical data, the next step was to estimate the associated coral mortality. The maximum bleaching mortality was related to DHW using another sigmoid curve (Gompertz function) fitted to data from the 2016 mass bleaching event on the GBR [3] (equation (2.13), figure 4d). These mortality curves are separated by differences in natural thermal tolerance of coral groups (measured in DHWs), with slow-growing corals tending to be more thermally tolerant than fast-growing corals (equation (2.14), assumption 3b) [91,92].

The process for setting bleaching mortality in future years using the relationships described above was as follows:

(i) DHW was set by randomly sampling from beneath the maximum DHW curve shown in figure 4b.
(ii) The geographical radius of the heatwave zone was set by again sampling randomly from beneath the maximum DHW curve and then scaling by the constant used to generate figure 4c.
(iii) For reefs within the heatwave zone, the mortality of each coral group was estimated by randomly selecting from beneath the square of the distribution for that group in figure 4d (i.e. $\chi^2$ distribution with one degree of freedom) and taking the square root. This last step weighted the mortality distribution towards higher values as suggested by the 2016 data (figure 4d).

This stochastic approach resulted in patchy distributions of bleaching mortality [3], which are more realistic than distributions that might be generated by a more deterministic relationship between DHW and coral mortality.

## 2.6. Ocean acidification

Corals build their exoskeleton with aragonite, and ocean acidification is lowering the aragonite saturation state of seawater. This process was represented as a broad-scale reduction in coral growth rates informed by recent field and laboratory experiments on reductions in calcification rates under decreasing ocean pH levels [48,93]. Observations of coral distributions in naturally low pH environments [94] and controlled laboratory experiments [95] both suggest that faster-growing corals may be more susceptible to acidification. Biogeochemical modelling further indicates that aragonite saturation rate on outer GBR reefs is on average 0.76 times that on inner GBR reefs [96]. This difference is comparable to the expected change in aragonite saturation rate over the next century assuming that recent rates of decline continue (0.76% per year [48]).

Coral responses to ocean acidification were, therefore, assumed to be dependent on their underlying growth rate, offshore location and the climate scenario (equation (2.16), figure 4e, assumption 3d). The formulation captures the decline in coral growth rates over recent decades evident in both laboratory [59] and field results [22,48]. By 2070, modelled coral growth rates fall to 56% below pre-industrial levels for RCP 4.5 or 84% below pre-industrial levels for RCP 8.5. These values are comparable to estimates of approximately 50% and greater than 100% derived from laboratory results [59]. The model formulation is also consistent with annual declines of 0.75–1.23% suggested by analyses of the skeletal structure of corals in the GBR [48,97], the Indo-Pacific [98] and Central America [99].

CoTS may also be influenced by ocean acidification. However, laboratory studies suggest that the effect is negative for larvae [49] and positive for juveniles [50], and may be further confounded by temperature dependencies. Effects over their life history are, therefore, highly uncertain and have been assumed to be negligible in the current model (assumption 3d).

## 2.7. Natural adaptation of corals

The thermal tolerance of any coral group could change through natural adaptation. There are various approaches that can be used to model this process and the rates and maximum extent of adaptation are still largely unknown [100]. We, therefore, implemented a parsimonious model that captured only the essential dynamics of coral adaptation with trade-offs between the key traits of thermal tolerance and growth rate. Agent-based models are well structured to capture such processes [101], in this instance, tracking key traits at the scale of individual reefs.

Following each bleaching event, the thermal tolerance of surviving corals (measured in DHWs) was increased by a factor that rose with both their inherent adaptability and the mortality rate associated with the event (assumption 4a). The rate of adaptation was limited by the adaptability parameter (equation

(2.15), figure 4*f*), which was assumed to be the same for all coral groups. However, groups more susceptible to bleaching had higher mortality and this selective pressure drove more rapid adaptation. This was considered the simplest conceivable model in which thermal tolerance increased with bleaching mortality, but remained unchanged if either adaptability or bleaching mortality were zero.

In the absence of continuing thermal stress, thermal tolerance gradually declined again as the community structure within each coral group recovered [102–104] or corals shuffled their zooxanthellae populations to more thermally tolerant symbiont types [72,103]. The exponential timescale for decline associated with a coral group's community structure was assumed to be inversely proportional to the growth rate of the group (ranging from 10 years for the fastest-growing corals to 100 years for the slowest growing corals) [72,102–104] (assumption 4a). However, it is acknowledged that shorter timescales (less than 5 years) may be appropriate where corals adapt by shuffling their zooxanthellae populations [72,103].

The adaptive capacity of corals was also limited by imposing both a cap on the cumulative change in thermal tolerance and a growth rate penalty per DHW increase in thermal tolerance. Empirical evidence suggests that even a single type of zooxanthellae can adapt their thermal tolerance by more than 3 DHW [105]. Considering the greater potential offered by shuffling zooxanthellae, a cap of 12 DHW on changes in thermal tolerance is not unreasonable (also compare with figure 4*b*). In any case, over 50-year projections, the growth rate penalty (assumed to be 1.0% per DHW of thermal tolerance) usually limited adaptation of populations below this cap. Specifically, to reach the adaptation cap, corals needed to be exposed to three to five successive bleaching events without any extended recovery periods. In the absence of continuing thermal stress, coral growth rates recovered commensurate with the decline in thermal tolerance.

Thermal tolerance was heritable in that recruitment from neighbouring reefs contributed to the average thermal tolerance of the receiving reef. However, averaging at the reef scale limited the propagation of traits, except to reefs where the existing coral cover was very low. An implicit model assumption was, therefore, that local adaptation in direct response to heat stress tended to be the main driver of adaptation (over the 50-year projection), rather than propagation of traits from reef to reef and across latitudes (i.e. genetic rescue) [106,107] (assumption 4b). This assumption has not yet been tested empirically and could potentially lead to overly pessimistic adaptation scenarios.

The net rate of adaptation in the model was largely controlled by the adaptability parameter (figure 4*f*). While adaptation rates on the GBR are largely unknown [72,100], setting adaptability to 5 delayed coral decline by around 10 years under RCP 4.5 and RCP 8.5, and longer under RCP 2.6. These effects are consistent with mid-range adaptive responses to sea surface temperature changes over the past two decades [51] as well as future projections [51,72]. We refer to this as a plausible level of natural adaption to emphasize the uncertainties associated with predicting natural adaptation.

## 2.8. Calibration against historical data

Model parametrizations have previously been calibrated for coral and CoTS population dynamics at the scale of individual reefs [28,56] and smaller networks of reefs [17]. This was extended here to cover the entire GBR system by comparing model ensemble statistics with historical coral cover in the northern, central and southern regions of the GBR estimated from the LTMP [55] (figure 5). Because CoTS outbreaks were an emergent behaviour in the model, their magnitude and timing could vary across ensemble members. However, outbreak characteristics such as the frequency of outbreaks and their propagation speed were compared with behaviours produced in individual model ensemble runs.

## 2.9. Interventions

Short- and long-term intervention options were identified from the *Great Barrier Reef Blueprint for Resilience* (http://elibrary.gbrmpa.gov.au/jspui/handle/11017/3287), the *Reef Restoration and Adaptation Program* (https://www.gbrrestoration.org/interventions) and from existing management practices. Interventions could be classified as either regional scale or reef scale. In the latter case, the number of reefs treated was generally limited by some form of intervention capacity. Under these circumstances, reefs designated as high priority were treated first. A total of 289 reefs were designated as high priority by the Great Barrier Reef Marine Park Authority based on factors such as their importance as tourism sites or their potential to contribute to recruitment of coral or CoTS [30–33].

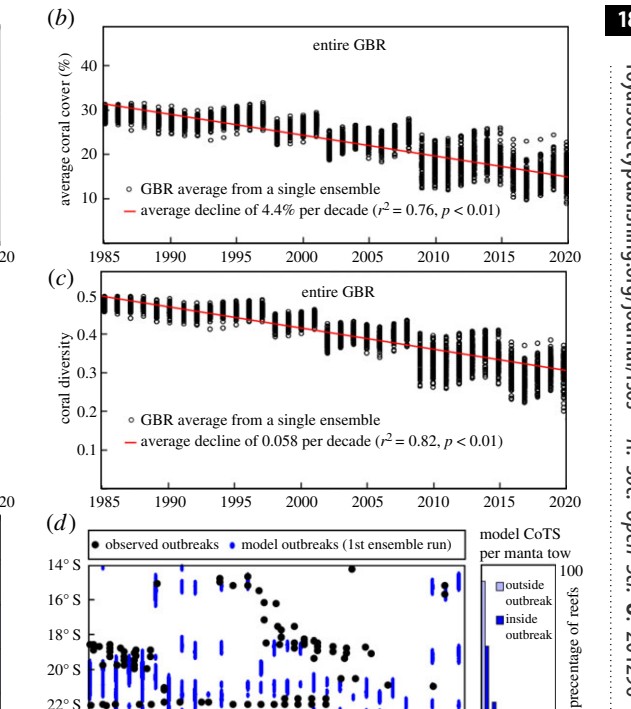

**Figure 5.** (*a*) Comparison of observed and modelled coral cover averaged over northern, central and southern reefs for the period 1986–2019. Observations are from the AIMS LTMP [55] covering 6–8% of GBR reefs in any year and represented here by the mean (red line) and 95% credible intervals (red shading). The model results are represented by the 100-member ensemble mean (blue dashed line) and ±2 s.d. spanning approximately 95% of the data in any year (blue shading). (*b*) Modelled annual coral cover averaged over all GBR reefs for the period 1985–2020 from all 100 ensemble runs. (*c*) As in (*b*) for modelled coral diversity (evenness index). (*d*) Comparison of observed latitudes of CoTS active outbreaks (greater than 1.0 CoTS per manta tow, equivalent to 67 CoTS ha$^{-1}$) [1] and model outbreak latitudes from the first model ensemble member. A histogram of average modelled CoTS density both outside of the outbreak zone and inside of the outbreak zone across the 100-member ensemble is shown in the right-hand panel.

## 2.10. Reducing the impact of flood plumes

Only a few per cent of reefs on the GBR are directly exposed to flood plumes containing elevated nutrients and suspended sediments [108] and quantifying the influence of catchment restoration on reef ecology is still a major challenge [10]. The model, therefore, focused on the potential influences of flood plumes on coral growth and recruitment of CoTS larvae. Reductions in these impacts (through improvements in coastal catchments) were assumed to asymptote towards an improved state over a timeframe of 20 years, which may be optimistic [10] (assumption 5a). The maximum achievable improvement was assumed to have an effect equivalent to reducing the intensity of tropical cyclones and lows by one cyclone category. For a category 3 cyclone, this had the effect of reducing the offshore scale of catchment influences by one-third. This equates to around 42% of the difference between southern and far northern catchments on the GBR, which has previously been used as an indication of the maximum improvement that might be achievable through catchment restoration [52]. Hence, the limits placed on catchment restoration in the model were broadly consistent with geographical differences in catchment condition.

## 2.11. Controlling CoTS

The implementation of baseline CoTS control in the model closely followed the approach used by control vessels currently operating on the GBR [14,19] (assumption 6a). Control firstly targeted priority reefs and then moved onto other reefs as allowed by vessel capacity. There were five vessels, each carrying eight divers. Every vessel conducted 20 voyages per year, each lasting 10 days over which 36 dive sessions were completed (i.e. average of four per day with 1 day lost in transiting). This gave a total of $5 \times 8 \times 20 \times 36 = 28$

800 dives per year distributed across the five vessels. Each dive took place at a single site within a reef, with the size of each site equating to the area that could be covered by a single dive under average CoTS densities (approx. 500 m long and 200 m wide or 10 ha).

CoTS were killed sequentially at each dive site until numbers fell below an ecological threshold that explicitly accounted for coral cover. Recent studies [109] suggest that even very low coral cover can support a detectable concentration of four CoTS ha$^{-1}$, whereas 30% coral cover can support 10 CoTS ha$^{-1}$ consistent with early estimates [26]. While there is evidence that very high coral cover may be able to support up to 28 CoTS ha$^{-1}$ [109], it is desirable to reduce CoTS densities significantly below the threshold for high coral cover where CoTS fertilization success may be enhanced [110]. A relatively conservative threshold can, therefore, be represented by a simple linear dependence on coral cover (equation (2.17)).

When reducing CoTS populations to the ecological threshold, it was assumed that younger CoTS were harder to detect, and, therefore, a smaller proportion of them were controlled (assumption 6b). Hence, after culling, age classes 2, 3, 4 and 5+ years were left with the respective average abundances of 0.63, 0.29, 0.065 and 0.015 times the threshold [53]. Any age class already below this abundance was not affected by the control programme.

The number of individual 40 min dives per site required to reduce CoTS below the ecological threshold has been estimated using empirical data from historical and recent control efforts [111] (equation (2.18)). Once a site had been reduced below the ecological threshold, control activities moved to the next dive site (or next reef if all sites had been treated). This process continued until all dives available for that year had been fully used.

## 2.12. Stabilizing coral rubble

Coral rubble was generated as a direct consequence of coral mortality during cyclone and bleaching events. The area of rubble cover was set at twice the contributing live coral cover (corresponding to a hemispherical surface of live coral collapsing onto a flat seabed). Rubble cover was tracked on all reefs and coral recruitment was prevented over the proportion of a reef covered by rubble. The rubble decayed exponentially with a decay timescale of 5.5 years consistent with empirical data on natural rubble consolidation [112].

Stabilization interventions can bond, mesh or remove rubble, all of which were modelled by reducing rubble limitations on coral recruitment, proportionate to the area stabilized. It was targeted only at priority reefs with low coral cover (less than 20%) and large areas of rubble (greater than 1 ha). Reefs satisfying these criteria were selected at random and a prescribed area of 1 ha of rubble was stabilized each year with the effort distributed over a maximum of 100 reefs. While the annual stabilization area was fixed to limit logistical requirements, it is clearly much larger than has any existing stabilization programme (assumption 7a).

## 2.13. Shading to reduce coral bleaching

Shading was specified in the model as a fixed reduction in DHW leading to reduced bleaching mortality (equation (2.13)). This simple parametrization implicitly represents reductions in both water temperatures and irradiance levels that typically contribute to bleaching. Using surface films or other shading devices may be effective in reducing heat stress on a limited number of reefs. We have, therefore, assumed a reduction of 12 DHW, while acknowledging potential limitations associated with warmer waters flowing onto the reef from outside the shading area (assumption 8a).

When solar radiation management is applied at regional or GBR-wide scales (sky brightening, cloud-brightening or ocean surface albedo modification [54,113]) or as global geoengineering (stratospheric aerosols [78]), more uniform reductions in heat stress may be achievable. For example, modelling the effect of radiative forcing on ocean temperatures over GBR reefs indicates that a 30% increase in low-level cloud albedo (corresponding to a 6.5% increase in average albedo) would have reduced heat stress by 7.5 ± 3.5 DHW over the summer of 2015–2016 and 8.3 ± 3.7 DHW over the summer of 2016–2017 (increasing sky albedo by 6.8% showed very similar benefits) [54]. These values were described as 'representative of a reasonable (perhaps aspirational) target for solar radiation management' and early field trials of a delivery system broadly support this view (https://www.theguardian.com/environment/2020/apr/17/scientists-trial-cloud-brightening-equipment-to-shade-and-cool-great-barrier-reef). While this implies a relatively positive outlook for solar radiation management, the research is still in an early phase and we have, therefore, adopted a more conservative reduction of 4.0 DHW across the GBR (assumption 8b).

## 2.14. Introducing thermally tolerant corals

A thermally tolerant coral group (strain) was characterized by lower rates of mortality during bleaching events (figure 4d). On the GBR, such corals could be seeded as larvae or outplanted as juveniles. High mortality expected immediately following seeding or outplanting was not explicitly modelled, so that the initial coverage represented successful introductions only. The total annual successful seeding or outplanting area of coral cover was fixed at 10 ha and distributed evenly across all priority reefs with low existing coral cover (less than 20%). This is clearly a much larger programme than has ever been demonstrated in the field (assumption 9a).

Thermally tolerant corals were assumed to be capable of interbreeding with some fraction of the existing coral community (assumption 9b). Because each of the other groups represented many coral species, interbreeding was limited to a specified proportion of just one coral group (staghorn *Acropora*). Hybrids recruited to a reef were allocated proportionally to each of the two interbreeding groups, with proportionally weighted changes to their thermal tolerance. While this provides a reasonable starting point for the modelling, an extensive research breeding programme will clearly be required to properly quantify these processes.

## 2.15. Mitigating ocean acidification

Declines in growth rates of corals due to ocean acidification may be offset by releasing alkaline chemicals [22] or growing plants such as macroalgae around reefs [23]. In the model, such interventions were represented by protecting all priority reefs completely from the effects of ocean acidification (equation (2.16)). This represents a large intervention programme with a very high level of effectiveness.

## 2.16. Ensemble runs testing interventions

Interventions could be tested individually or in combination under any specified RCP scenario. Each simulation started in 1950 and ended in 2070, with the first 30 years treated as an equilibration period. Because the model and model forcing included a large number of stochastic elements, every run was repeated 100 times allowing ensemble statistics to be calculated (i.e. 100-member ensembles). Ensemble means and variances were found to be insensitive to ensemble size for more than 25 members.

Historical cyclones and severe heatwaves (1950–2019) were applied as localized coral mortality events, dependent on cyclone intensity and DHW, respectively, switching to stochastic events (2020–2070) described by probabilistic frequency and intensity distributions based on literature projections. This approach provided statistically representative ensembles. After comparing model estimates with historical coral cover data, we modelled coral trajectories under three climate scenarios (Representative Concentration Pathways RCP 2.6, RCP 4.5 and RCP 8.5). We then included interventions (table 1) by comparing responses under RCP 4.5, which has been used previously in the context of global interventions [78] and is very similar to RCP 6.0 in terms of ocean heating to 2070 [72].

For each run within an ensemble, the initial cover of each of the five coral groups was varied randomly between 0 and 10%, giving an average total coral cover of 25%. Adult CoTS populations were varied randomly between 0 and 0.75 CoTS per manta tow (0–50 CoTS ha$^{-1}$). The percentage cover of each coral group and the concentration of each age class of CoTS were then recorded at every reef in every year within the ensemble. These data could be used to estimate the corresponding total coral cover (equation (2.7)) and coral diversity (equation (2.8)). The latter was specified in terms of the group evenness index (i.e. normalized Shannon's entropy) [114]. When only one coral group was present, the diversity was $\mathcal{J} = 0$, and when all groups had equal coverage, the diversity was $\mathcal{J} = 1$. The recorded model outputs were used to calculate time-series of ensemble averages and percentiles for coral cover, coral diversity and CoTS density for each region and over the entire GBR.

The effectiveness of interventions was assessed by comparing forecast projections with and without intervention for total coral cover as a percentage of available reef habitat area, averaged across all GBR reefs. To account for underlying variability across the model ensemble, the effect size of any change in coral cover associated with interventions was estimated in terms of Cohen's $d$ [115] (equation (2.19)). This annual statistic compared the absolute difference in ensemble-averaged coral cover with and without intervention, relative to the underlying variability across the ensemble. While alternative metrics are available (e.g. the $t$-statistic), Cohen's $d$ has the advantage that it quantifies the size of the difference and is associated with broadly accepted descriptors for magnitude of the difference (small,

medium or large). In this context, $p$-values are not meaningful as they decrease with the size of the model ensemble [116] and, therefore, have not been presented.

## 3. Results

Comparison of model ensemble statistics with historical coral cover estimated from the LTMP [55] shows a high level of consistency across the northern, central and southern regions of the GBR (figure 5a). From 1985 to 2020, GBR coral cover in the model declined from near 30% to 15% of total available habitat (figure 5b). This corresponds to an average loss rate of 4.4% per decade ($r^2 = 0.76$, $p < 0.01$). Over the same period, coral diversity (as measured by the evenness index) declined from near 0.50 to 0.30 (figure 5c), corresponding to a rate of 0.058 per decade ($r^2 = 0.82$, $p < 0.01$). A sharp decline in both coral cover and coral diversity in 2009 was followed by almost a doubling in the range of ensemble results that was then sustained until 2020 (figure 5b,c). This switch to a more uncertain trajectory appears to have been triggered by tropical cyclone Hamish, which unlike most cyclones tracked parallel to the coast and thereby impacted a large proportion of the GBR.

Because CoTS outbreaks were an emergent behaviour in the model, their magnitude and timing could vary across ensemble members. However, they generally showed southward movement, propagating relatively rapidly from around 15° S to 17° S and then more gradually from 17° S to 20° S (figure 5d). This is broadly consistent with the observed patterns [1], with outbreaks propagating approximately 1° south every 3 years from 17° S to 20° S [1,117,118], although it has been argued that the evidence for simple wave-like propagation is equivocal [119]. Both observations and model results also suggest that outbreaks between 20° S and 22.5° S tend to be locally more persistent (chronic) and largely independent of those to the north (figure 5d). These north–south differences are probably associated with reef geography and oceanographic connectivity patterns [120].

In the absence of interventions, all climate scenarios showed an ongoing decline in GBR coral cover from 2020 to 2070 (figure 6a). However, because future cyclones and heatwaves were controlled by probability distributions (rather than historical patterns), ensembles spanned a wide range of outcomes (figure 6b). For RCP 4.5, the average annual fall in GBR coral cover was 0.27% ($r^2 = 0.54$, $p < 0.01$). While this decline is less steep than the historical decline (1985–2019), annual losses as a fraction of remaining coral cover increased. Starting from an ensemble average coral cover of approximately 16% in 2019, the three warming scenarios diverged from 2035, and by 2070 average coral cover was approximately 6% for RCP 2.6, 3% for RCP 4.5 and 1.5% for RCP 8.5 (figure 6a). Including a plausible (but uncertain) level of natural adaptation delayed coral decline by a decade under RCP 4.5 and RCP 8.5, and longer under RCP 2.6, consistent with reported mid-range adaptive responses to SST projections [72].

Maintaining existing intervention commitments (reduced flood plume impacts and five CoTS control vessels) was effective in reducing average CoTS densities throughout the projection period (figure 6e). However, improvements in average coral cover were modest and only occurred after 2040 (figure 6b,e) as the impacts of flood plumes were gradually reduced.

When applied individually, most intervention strategies were ineffective over the projection period (figure 6c), particularly those limited in spatial coverage (e.g. stabilizing substrate or moderating ocean acidification on priority reefs). The notable exception was regional shading capable of reducing the DHW exposure of reefs. This intervention benefited coral cover despite also supporting higher CoTS densities to 2040 (figure 6e). Interestingly, introduction of thermally tolerant corals had a negative impact on coral cover until 2050 (figure 6c). This outcome was due to the more consistent coral availability supporting increased CoTS densities (figure 6e).

Interventions were more effective when combined (figure 6d). In combination with either enhanced CoTS control or thermally tolerant corals, regional shading produced improvements in coral cover throughout the projection period (figure 6e). However, the combination of enhanced CoTS control and thermally tolerant corals without shading was less successful. Combining all three interventions was by far the most effective (figure 6d) and by 2070 had 24% higher coral cover than the highest dual strategy combination and 53% higher coral cover than without intervention. Natural adaptation of corals has potential to enhance these gains (cf. assumptions 3a and 3d), although adaptation rates remain uncertain.

## 4. Discussion and conclusion

The GBR is a large complex ecosystem and the CoCoNet model only included the 'minimum realistic' components and feedbacks necessary to capture responses from single-reef [28] to whole-of-GBR

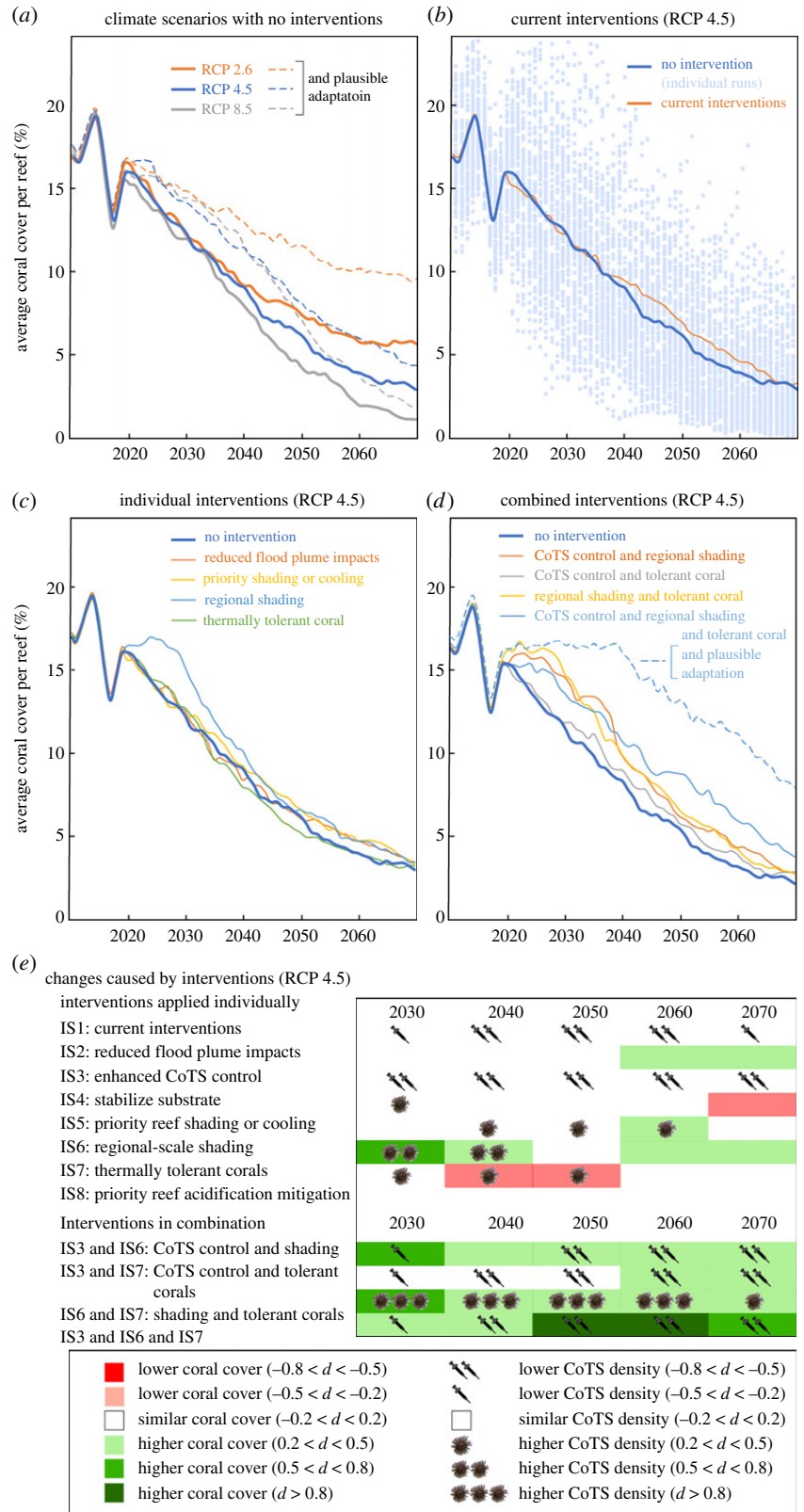

**Figure 6.** Model coral cover averaged across all reefs and 100 ensemble members: (*a*) three climate projections with no intervention, with and without plausible levels of natural adaptation of corals to thermal stress; (*b*) current interventions (including data from individual ensemble runs) compared with no intervention; (*c*) interventions applied individually under RCP 4.5 (excluding those that had only a small effect on coral cover prior to 2070); (*d*) combination of interventions under RCP 4.5 including one combination with a plausible level of natural adaptation of corals to thermal stress and (*e*) effects of interventions on coral cover and CoTS density under RCP 4.5 for years 2030, 2040, 2050, 2060 and 2070. Cohen's *d* is a measure of effect size (small when |*d*| < 0.2; small to medium when 0.2 < |*d*| < 0.5; medium to large when 0.5 < |*d*| < 0.8 and large when |*d*| > 0.8).

scales [17]. Even with such minimal complexity, some aspects of the model remain to be explored in detail (e.g. responses in coral community composition to interventions). While there are no technical impediments to adding additional components and processes to the model, increasing complexity can increase errors and reduce the relevance and usefulness of models [121]. Even our parsimonious approach requires a significant number of assumptions (table 2), some of which continue to be contested. Acknowledging that complex system models can never be fully verified [122], the formulation has been validated quantitatively by comparing outputs with available empirical data (i.e. positivism perspective [123]), as well as qualitatively in terms of its fitness for purpose through continuous evaluation by the RRAP group of experts (https://www.gbrrestoration.org/about-us) (i.e. relativism perspective [123]).

There are varying levels of uncertainty associated with modelling interventions. For example, the complex series of hypothesized biophysical interactions connecting flood plumes to the responses of coral and CoTS [9,10]. There are also large uncertainties associated with the climate scenarios and their ecological responses (figure 6b), which are based on limited empirical data from a small number of historical events. We have attempted to capture these uncertainties within our ensemble modelling approach, and thereby provide insights into possible trajectories of the GBR over the next 50 years. While there may be irreducible uncertainties inherent in the projections, the guidance they provide in relation to the relative performance of intervention options should be relatively robust.

Ensemble-averaged coral cover declined across the GBR from 1985 to 2020 by 4.4% per decade (figure 5b). A more rapid decline from 1996 to 2017 of 6.1% per decade ($r^2 = 0.67$, $p < 0.01$) is similar to the recent estimate of 6.7% obtained by fitting gridded coral cover to disturbance history south of $14°$S [2]. This period included tropical cyclone Hamish in 2009, which impacted an anomalously large area of the GBR by moving parallel to the coast. This event appears to have triggered a change in the state of the modelled GBR, characterized by more depleted coral cover and diversity, low CoTS densities and less certain population trajectories (figure 5b,c). Similar changes are evident in the LTMP data (figure 5a,d), potentially representing an ecological tipping point for the GBR system [56] after which average coral cover is around 15%.

All future projections suggest continued decline, with an ensemble-averaged rate of 0.27% per year for RCP 4.5. However, because future cyclones and heatwaves were specified in terms of probability distributions (figure 4a–c), a wide range of trajectories were possible. For example, individual model ensembles suggest that without intervention, coral cover in 2050 could range from a relatively healthy 16% to a catastrophic 1% (figure 6b), although the probability of such extremes is very low. This underlines the need to assess modelled interventions in terms of relative risk and recognize that no intervention can guarantee a healthy reef system in a changing climate.

The potential value of interventions is in slowing the rate of decline and allowing time for development of other more effective interventions, natural adaptation and, ultimately most important, global climate action [8]. The model results suggest that delays of one to two decades are likely to be feasible, and this might be further extended by better targeting interventions (spatially or temporally) or enhancing their effectiveness in novel ways. In any case, two decades may be sufficient to help evade tipping points leading to ecological collapse [124] or, at the very least, allow for social and economic adaptation to cope with the changed state [125,126].

Key lessons relating to the relative effectiveness of interventions have emerged from our model runs. Firstly, interventions limited to a relatively small proportion of reefs were largely ineffective at the whole-of-GBR scale (figure 6c,e). These included stabilization of coral rubble, local shading and protection from ocean acidification, each of which focused on priority reefs comprising only 7.7% of GBR reefs. River plumes directly impact an even smaller percentage of GBR reefs [127], placing limits on the potential effectiveness of catchment restoration [11] (but recognizing that water quality improvements provide key benefits to other parts of the ecosystem). This is not to say that localized strategies are not effective with respect to the reefs that are treated. For example, by 2040, shaded priority reefs on average had double the coral cover of those without shading (not shown).

Secondly, by reducing water temperatures, irradiance levels and associated bleaching [54], regional shading appears to be particularly effective in helping to maintain coral cover and was the only single intervention to be effective over the next two decades (figure 6c–e). This is not surprising, given that increased coral mortality over the next 50 years is expected to be primarily associated with increased frequency and severity of heatwaves causing mass bleaching events [3,4,6]. It should, however, be cautioned that technologies capable of reducing upper ocean heating at the scenario rate (4 DHW) and spatial scale (entire GBR) are yet to be demonstrated, and their social and legal acceptability is also untested [113].

Thirdly, any intervention that focused only on protecting or restoring corals also benefited CoTS through enhanced food supply [17]. Most notably, the persistence of thermally tolerant coral helped maintain the connectivity of highly fecund CoTS populations within the reef network [31], resulting in total coral cover falling below levels without intervention (figure 6c,e). This feedback was eliminated and coral cover improved when CoTS control was included as part of the intervention strategy [31] (figure 6d,e). A clear lesson is that CoTS control must form part of any successful coral restoration strategy on the GBR.

While we applied interventions from 2020 to 2070, some interventions may not be logistically, technologically or socially feasible to implement on short timescales. Furthermore, their relative effectiveness may vary greatly over the future evolution of the GBR and from region to region. Future work will consider when and where interventions should be optimally deployed as part of a broader recovery and adaptation strategy. For example, our results suggest that regional shading in combination with CoTS control will be relatively effective until 2040, at which time thermally tolerant coral could begin to play an important role in slowing the decline of the GBR. However, constraining the costs of such interventions will require deployments to be optimized in both space and time.

Another key question for the future of the GBR is the rate of natural thermal adaptation within coral communities. We found that plausible levels of natural adaptation [72] may delay coral decline by a decade, or even two decades when supported by a combination of interventions. However, rates of adaptation are highly uncertain and improving our understanding and modelling of these processes is a high priority for ongoing research.

Data accessibility. The CoCoNet model code and user interface is provided as electronic supplementary material (https://doi.org/10.5061/dryad.zkh18937s) and can be run within the NetLogo environment (v. 6.0.4), which can be downloaded free of charge at https://ccl.northwestern.edu/netlogo/6.0.4/.

Authors' contributions. All authors contributed to the study design and writing. In addition, S.A.C. developed and ran the model; K.R.N.A. contributed to model parametrization of bleaching; R.C.B., C.S.F. and D.A.W. contributed to model parametrization of CoTS control; M.E.B. contributed to the parametrization of flows and flood plumes; D.H. contributed to the parametrization of solar radiation management; R.B. contributed to the prioritization of interventions and provided other advice from a policy and management perspective; R.G. calculated the reef connectivity used in the model; A.J.H. contributed to the prioritization of interventions; and E.E.P. developed key ecological model components.

Competing interests. We declare we have no competing interests.

Funding. This research was partially funded by the Reef Restoration and Adaptation Program (RRAP, https://gbrrestoration.org) and the Australian Government's National Environmental Science Program (NESP).

Acknowledgements. Our thanks to all members of the RRAP Steering Committee and RRAP Core Team for their expert advice throughout the study. This work is part of a collaboration with Peter Mumby and Yves-Marie Bozec from the University of Queensland, whose results using the model Reefmod will be reportedly separately.

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
