## [Peer Review File · Royal Society Open Science]

Review History

RSOS-201296.R0 (Original submission)

Review form: Reviewer 1

Is the manuscript scientifically sound in its present form?

Yes

Are the interpretations and conclusions justified by the results?

Yes

Is the language acceptable?

Yes

Do you have any ethical concerns with this paper?

No

Have you any concerns about statistical analyses in this paper?

No

Recommendation?

Accept with minor revision (please list in comments)

Comments to the Author(s)

Condie et al. present an extensive modeling effort to explore the potential benefits of large-scale intervention efforts to prevent coral decline on the Great Barrier Reef. I commend the authors on a thorough and accessible description of the many parameters and assumptions incorporated in their model. Overall, I found the conclusions to be well supported by the data and the findings of high value to the coral reef management community. In particular, the authors did an excellent job of distilling the core conclusions from their work as key recommendations for intervention planning.

The paper is well written and effectively organized. My only major comment is that the paper would benefit from a more thoughtful discussion of the limitations of their model assumptions and how their interpretation of the data might change if their assumptions varied slightly. I see at least two points worth elaborating on:

First, coral adaptation is incorporated in the model but it does not seem like changes in coral community composition towards more resilient (or less tasty to COTs) species have been considered. There are well documented changes in species composition on the GBR, which are likely to influence future coral responses to thermal stress, cyclones, and COTs. How might the projections of coral cover change if a shift in coral composition reduced the amount of food available to COTs, decreased bleaching mortality, and reduced the extent of cyclone damage? This seems like a plausible transformation with widespread impacts on model outputs and warrants some consideration.

Second, the concept of shading is put forth as an important intervention strategy with potentially important benefits to corals. The authors briefly acknowledge that the feasibility of this is unknown but it seems reasonable to assume that it is quite literally impossible. Given that, it seems pertinent the authors consider how the strong effects of shading may be influenced by model assumptions. A reduction of 4 DHW seems generous as this is $\frac{1}{4}$ to $\frac{1}{3}$ of the heat experienced by the GBR in 2015/16. If the benefits of shading weren't quite so large, would this intervention still be considered one of the most critical? It seems like some examination of the sensitivity of model output to the shading assumption would help contextualize how seriously this intervention should be considered.

I have provided a few minor comments below to help improve clarity.

Minor comments:

Line 226 – heat waves occasionally occurs as 2 words. Make sure to be consistent with this term as it is predominantly used, “heatwave”

Lines 497-503 – this is slightly confusing with table 1 that shows shading and cooling (?) as equivalent to reductions of 12 dhw. It seems like IS5 from table one is not included in the model while IS 6 is? Relative to 12, a reduction by 4 dhw seems small which doesn't follow well from the sentence in lines 498-501 describing “much higher dhw reductions”

I suggest revising this section for clarity, specifying exactly which mitigation measures are being estimated to produce the 4 dhw reduction, and reiterating that while identified in table 1, cooling is not incorporate in the model as an intervention.

This section also poorly characterizes the distinction between regional shading and priority shading/cooling illustrated in figure 6.

Line 666 – there is a comma missing after i.e.

Line 718 – with such a concrete example of local management improvements it seems like this would be worth including in the supplement rather than just stating it is “not shown”

Lines 719-726 – this discussion seems to be linked more directly to heating rather than shading. A brief description about why shading is useful seems appropriate. Is it the reduction in irradiance stress during a heatwave or does shading quantifiably reduce temperatures? Given water flow I suspect the former but it is written as though the latter is most likely.

Line 723 – another instance of heat wave being 2 words.

Decision letter (RSOS-201296.R0)

This year has been very difficult for everyone, and we want to take the opportunity to thank you for your continued support in 2020.

The Royal Society Open Science editorial office will be closed from the evening of Friday 18 December 2020 until Monday 4 January 2021. We will not be responding during this time. If you have received a deadline within this time period, please contact us as soon as possible to allow us to extend the deadline. If you receive any automated messages during this time asking you to meet a deadline, we offer apologies and invite you to respond after the festive period or during normal working hours.

With our best for a peaceful festive period and New Year, and we look forward to working with you in 2021.

Dear Dr Condie

The Editors assigned to your paper RSOS-201296 "Large-scale interventions may delay decline of the Great Barrier Reef" have now received comments from reviewers and would like you to revise the paper in accordance with the reviewer comments and any comments from the Editors. Please note this decision does not guarantee eventual acceptance.

Please submit your revised manuscript and required files (see below) no later than 21 days from today's (ie 18-Dec-2020) date. Note: the ScholarOne system will 'lock' if submission of the revision is attempted 21 or more days after the deadline. If you do not think you will be able to meet this deadline please contact the editorial office immediately.

on behalf of Dr Melita Samoily (Associate Editor) and Pete Smith (Subject Editor)
openscience@royalsociety.org

Associate Editor Comments to Author (Dr Melita Samoily):

Please accept our apologies for the delay in issuing a decision. It has been unusually difficult to secure even one report on your paper - indeed, the long delay to get to this point, and the comprehensive report provided by the reviewer motivates the editors to make a recommendation on one report. While the reviewer is broadly positively inclined towards your paper, we are issuing a revision to encourage you to fully address their concerns - it is likely that the paper will be returned to this referee for a further assessment after submission of the revision, unless the editors are persuaded your changes obviate the need for this step. We wish you every success, and all the best for the festive season.

Reviewer comments to Author:

Reviewer: 1
Comments to the Author(s)

Condie et al. present an extensive modeling effort to explore the potential benefits of large-scale intervention efforts to prevent coral decline on the Great Barrier Reef. I commend the authors on a thorough and accessible description of the many parameters and assumptions incorporated in their model. Overall, I found the conclusions to be well supported by the data and the findings of high value to the coral reef management community. In particular, the authors did an excellent job of distilling the core conclusions from their work as key recommendations for intervention planning.

The paper is well written and effectively organized. My only major comment is that the paper would benefit from a more thoughtful discussion of the limitations of their model assumptions and how their interpretation of the data might change if their assumptions varied slightly. I see at least two points worth elaborating on:

First, coral adaptation is incorporated in the model but it does not seem like changes in coral community composition towards more resilient (or less tasty to COTs) species have been considered. There are well documented changes in species composition on the GBR, which are likely to influence future coral responses to thermal stress, cyclones, and COTs. How might the projections of coral cover change if a shift in coral composition reduced the amount of food

available to COTs, decreased bleaching mortality, and reduced the extent of cyclone damage? This seems like a plausible transformation with widespread impacts on model outputs and warrants some consideration.

Second, the concept of shading is put forth as an important intervention strategy with potentially important benefits to corals. The authors briefly acknowledge that the feasibility of this is unknown but it seems reasonable to assume that it is quite literally impossible. Given that, it seems pertinent the authors consider how the strong effects of shading may be influenced by model assumptions. A reduction of 4 DHW seems generous as this is $\frac{1}{4}$ to $\frac{1}{3}$ of the heat experienced by the GBR in 2015/16. If the benefits of shading weren't quite so large, would this intervention still be considered one of the most critical? It seems like some examination of the sensitivity of model output to the shading assumption would help contextualize how seriously this intervention should be considered.

I have provided a few minor comments below to help improve clarity.

Minor comments:

Line 226 – heat waves occasionally occurs as 2 words. Make sure to be consistent with this term as it is predominantly used, “heatwave”

Lines 497-503 – this is slightly confusing with table 1 that shows shading and cooling (?) as equivalent to reductions of 12 dhw. It seems like IS5 from table one is not included in the model while IS 6 is? Relative to 12, a reduction by 4 dhw seems small which doesn't follow well from the sentence in lines 498-501 describing “much higher dhw reductions”

I suggest revising this section for clarity, specifying exactly which mitigation measures are being estimated to produce the 4 dhw reduction, and reiterating that while identified in table 1, cooling is not incorporate in the model as an intervention.

This section also poorly characterizes the distinction between regional shading and priority shading/cooling illustrated in figure 6.

Line 666 – there is a comma missing after i.e.

Line 718 – with such a concrete example of local management improvements it seems like this would be worth including in the supplement rather than just stating it is “not shown”

Lines 719-726 – this discussion seems to be linked more directly to heating rather than shading. A brief description about why shading is useful seems appropriate. Is it the reduction in irradiance stress during a heatwave or does shading quantifiably reduce temperatures? Given water flow I suspect the former but it is written as though the latter is most likely.

Line 723 – another instance of heat wave being 2 words.

===PREPARING YOUR MANUSCRIPT===

===PREPARING YOUR REVISION IN SCHOLARONE===

- If you are providing image files for potential cover images, please upload these at this step, and inform the editorial office you have done so. You must hold the copyright to any image provided.
- A copy of your point-by-point response to referees and Editors. This will expedite the preparation of your proof.

- Ensure that your data access statement meets the requirements at <https://royalsociety.org/journals/authors/author-guidelines/#data>. You should ensure that you cite the dataset in your reference list. If you have deposited data etc in the Dryad repository, please include both the 'For publication' link and 'For review' link at this stage.
- If you are requesting an article processing charge waiver, you must select the relevant waiver option (if requesting a discretionary waiver, the form should have been uploaded at Step 3 'File upload' above).
- If you have uploaded ESM files, please ensure you follow the guidance at <https://royalsociety.org/journals/authors/author-guidelines/#supplementary-material> to include a suitable title and informative caption. An example of appropriate titling and captioning may be found at https://figshare.com/articles/Table_S2_from_Is_there_a_trade-off_between_peak_performance_and_performance_breadth_across_temperatures_for_aerobic_scops_in_teleost_fishes_/3843624.

Author's Response to Decision Letter for (RSOS-201296.R0)

See Appendix A.

RSOS-201296.R1 (Revision)

Review form: Reviewer 1

Is the manuscript scientifically sound in its present form?

Yes

Are the interpretations and conclusions justified by the results?

Yes

Is the language acceptable?

Yes

Do you have any ethical concerns with this paper?

No

Have you any concerns about statistical analyses in this paper?

No

Recommendation?

Accept as is

Comments to the Author(s)

The authors can be commended on an extensive and well-executed study. I look forward to seeing this in print.

Decision letter (RSOS-201296.R1)

Dear Dr Condie,

It is a pleasure to accept your manuscript entitled "Large-scale interventions may delay decline of the Great Barrier Reef" in its current form for publication in Royal Society Open Science. The comments of the reviewer(s) who reviewed your manuscript are included at the foot of this letter.

on behalf of Dr Melita Samoilys (Associate Editor) and Pete Smith (Subject Editor)
openscience@royalsociety.org

Reviewer comments to Author:

Reviewer: 1

Comments to the Author(s)

The authors can be commended on an extensive and well-executed study. I look forward to seeing this in print.

Appendix A

Associate Editor Comments to Author (Dr Melita Samoily):

Please accept our apologies for the delay in issuing a decision. It has been unusually difficult to secure even one report on your paper - indeed, the long delay to get to this point, and the comprehensive report provided by the reviewer motivates the editors to make a recommendation on one report. While the reviewer is broadly positively inclined towards your paper, we are issuing a revision to encourage you to fully address their concerns - it is likely that the paper will be returned to this referee for a further assessment after submission of the revision, unless the editors are persuaded your changes obviate the need for this step. We wish you every success, and all the best for the festive season.

Reviewer comments to Author:

Reviewer: 1

Comments to the Author(s)

Condie et al. present an extensive modeling effort to explore the potential benefits of large-scale intervention efforts to prevent coral decline on the Great Barrier Reef. I commend the authors on a thorough and accessible description of the many parameters and assumptions incorporated in their model. Overall, I found the conclusions to be well supported by the data and the findings of high value to the coral reef management community. In particular, the authors did an excellent job of distilling the core conclusions from their work as key recommendations for intervention planning.

Our thanks for such positive encouragement.

The paper is well written and effectively organized. My only major comment is that the paper would benefit from a more thoughtful discussion of the limitations of their model assumptions and how their interpretation of the data might change if their assumptions varied slightly. I see at least two points worth elaborating on:

First, coral adaptation is incorporated in the model but it does not seem like changes in coral community composition towards more resilient (or less tasty to COTs) species have been considered. There are well documented changes in species composition on the GBR, which are likely to influence future coral responses to thermal stress, cyclones, and COTs. How might the projections of coral cover change if a shift in coral composition reduced the amount of food available to COTs, decreased bleaching mortality, and reduced the extent of cyclone damage? This seems like a plausible transformation with widespread impacts on model outputs and warrants some consideration.

All of the processes mentioned already operate in the model and the suggested scenario of shifting coral composition certainly occurs. Specifically, the model includes 5 coral groups (plus the thermally tolerant group) that all differ in their preference by CoTS and susceptibilities to cyclones and heat stress. The thermal tolerance of each coral group can also independently adapt in response to heatwave events. However, apart from the high-level view provided by the declining trend in coral diversity (evenness index for coral groups) presented in Figure 5, we have not attempted to explore trajectories of individual coral groups in this paper. While

certainly of interest, we feel that given the length and complexity of the current manuscript, these aspects should be explored in future publications. We have now noted this in the discussion: “Even with such minimal complexity, some aspects of the model remain to be explored in detail (e.g. responses in coral community composition to interventions).” In any case, we have now made it clearer that important effects associated with coral composition are included in the model. For example, we have now been more explicit in Section 3.1: “These groups nominally corresponded to staghorn *Acropora*, tabular *Acropora*, *Montipora*, *Poritidae* and *Favids*, distinguished within the model in terms of their growth rates, preference by CoTS, and susceptibility to environmental impacts such as cyclones and heat stress (Tables 3-5).” Differences between coral groups in terms of CoTS preferences and their responses to cyclones and heatwaves are further described in Sections 3.1, 3.4 (with Figure 4a) and 3.5 (with Figure 4d) respectively. The description of adaptation has also been expanded in Section 3.7: “The rate of adaptation was limited by the adaptability parameter (equation 15, Figure 4f), which was assumed to be the same for all coral groups. However, groups more susceptible to bleaching had higher mortality and this selective pressure drove more rapid adaptation.”

Second, the concept of shading is put forth as an important intervention strategy with potentially important benefits to corals. The authors briefly acknowledge that the feasibility of this is unknown but it seems reasonable to assume that it is quite literally impossible. Given that, it seems pertinent the authors consider how the strong effects of shading may be influenced by model assumptions. A reduction of 4 DHW seems generous as this is $\frac{1}{4}$ to $\frac{1}{3}$ of the heat experienced by the GBR in 2015/16. If the benefits of shading weren't quite so large, would this intervention still be considered one of the most critical? It seems like some examination of the sensitivity of model output to the shading assumption would help contextualize how seriously this intervention should be considered.

We would argue that 4 DHW is not unreasonable and certainly not impossible (with sufficient investment). While it may represent $\frac{1}{4}$ to $\frac{1}{3}$ of the DHW experienced by the GBR in 2015/16, it must be remembered that DHW is itself an anomaly relative to the daily summer climatology. According to (54) the average reduction in incoming shortwave radiation associated with a scenario corresponding to 8 DHW is only around 6.7%. We have been even more conservative in using 4 DHW. However, the reviewer's comment points to the need for more detailed justification of this value. In Section 3.13 we have therefore added the following: “For example, modelling the effect of radiative forcing on ocean temperatures over GBR reefs indicates that a 30% increase in low-level cloud albedo (corresponding to a 6.5% increase in average albedo) would have reduced heat stress by 7.5 ± 3.5 DHW over the summer of 2015-16 and 8.3 ± 3.7 DHW over the summer of 2016-17 (increasing sky albedo by 6.8% showed very similar benefits) (54). These values were described as “representative of a reasonable (perhaps aspirational) target for solar radiation management” and field trials of a prototype delivery system broadly support this view (<https://www.theguardian.com/environment/2020/apr/17/scientists-trial-cloud-brightening-equipment-to-shade-and-cool-great-barrier-reef>). While this implies a relatively positive outlook for solar radiation management, the research is still in an early phase and we have therefore adopted a more conservative reduction of 4.0 DHW across the GBR (assumption 8b).”

I have provided a few minor comments below to help improve clarity.
Minor comments:

Line 226 – heat waves occasionally occurs as 2 words. Make sure to be consistent with this term as it is predominantly used, “heatwave”

‘Heatwave’ has now been adopted throughout.

Lines 497-503 – this is slightly confusing with table 1 that shows shading and cooling (?) as equivalent to reductions of 12 dhw. It seems like IS5 from table one is not included in the model while IS 6 is? Relative to 12, a reduction by 4 dhw seems small which doesn’t follow well from the sentence in lines 498-501 describing “much higher dhw reductions”

I suggest revising this section for clarity, specifying exactly which mitigation measures are being estimated to produce the 4 dhw reduction, and reiterating that while identified in table 1, cooling is not incorporate in the model as an intervention.

Yes, this was confusing. We actually did undertake scenarios for both local shading (IS5: 12 DHW on selected reefs) and regional shading (IS6: 4 DHW across the GBR). We have expanded Table 2 to explicitly describe the 12 DHW assumption for local shading (now assumption 8a) and revised the associated paragraphs as follows:

“Shading was specified in the model as a fixed reduction in DHW leading to reduced bleaching mortality (equation 13). Utilising surface films or other shading devices may be effective in reducing heat stress on a limited number of reefs. We have therefore assumed a reduction of 12 DHW, while acknowledging potential limitations associated with warmer waters flowing onto the reef from outside the shading area (assumption 8a).”

And then: “While this implies a relatively positive outlook for solar radiation management, the research is still in an early phase and we have therefore adopted a more conservative reduction of 4.0 DHW across the GBR (assumption 8b).”

This section also poorly characterizes the distinction between regional shading and priority shading/cooling illustrated in figure 6.

The changes described in the previous point now provide a clear distinction between regional and local priority shading.

Line 666 – there is a comma missing after i.e.

We haven’t inserted a comma after “i.e.” anywhere in the manuscript, which seems to be consistent with published papers in RSOS.

Line 718 – with such a concrete example of local management improvements it seems like this would be worth including in the supplement rather than just stating it is “not shown”

The strength of the model and focus of the paper is on regional-scale responses. If we were to show local improvements for shading, then for consistency we would probably need to show it for all the interventions applied preferentially on priority reefs, which is most of them (CoTS control, rubble stabilisation, thermally tolerant corals, protection from acidification). There is also no real surprise that as bleaching becomes more frequent, locally excluding this impact will increase local coral cover. Hence, noting the effect in a quantitative way (i.e. double the coral cover in 2040) still seems to us to

capture the key point without distracting from the main focus of the paper.

Lines 719-726 – this discussion seems to be linked more directly to heating rather than shading. A brief description about why shading is useful seems appropriate. Is it the reduction in irradiance stress during a heatwave or does shading quantifiably reduce temperatures? Given water flow I suspect the former but it is written as though the latter is most likely.

The modelling treats the temperature and irradiance effects in combination. We agree that this was not very clear and have made changes in a number of sections of the manuscript. First, we have now been more explicit about the contributing factors in Section 3.5: “This exposure is expressed in terms of degree heating weeks (DHW), acknowledging that bleaching levels are usually influenced by a combination of water temperature and irradiance [89].” Second, in Section 3.13 we now note: “This simple parameterisation implicitly represents reductions in both water temperatures and irradiance levels that typically contribute to bleaching.” Third, in the discussion we now begin the paragraph referred to in the review by: “Secondly, by reducing water temperatures, irradiance levels and associated bleaching [54], regional shading appears to be particularly effective in helping to maintain coral cover ...”

Line 723 – another instance of heat wave being 2 words.

‘Heatwave’ has now been adopted throughout.

===PREPARING YOUR MANUSCRIPT===

Included as tracked changes.

Included after changes accepted.

All equation generated using the MS equation editor.

Please ensure that you include an acknowledgements' section before your reference list/bibliography. This should acknowledge anyone who assisted with your work, but

does not qualify as an author per the guidelines
at <https://royalsociety.org/journals/ethics-policies/openness/>.

Completed.

Vancouver selected in EndNote X8